# The association between limiting longstanding illness and serious psychological distress in adolescents: A secondary analysis of the UK Millennium Cohort Study

Gareth Martyn Palliser[1], Lorna K. Fraser[2], Kate E. Mooney[1,3], Stuart W. Jarvis[1]*

**1** Department of Health Sciences, University of York, York, United Kingdom, **2** Cicely Saunders Institute, King's College London, London, United Kingdom, **3** Better Start Bradford Innovation Hub, Bradford Institute for Health Research, Bradford, United Kingdom

\* stuart.jarvis@york.ac.uk

## Abstract

### Background

Previous studies have shown associations between specific limiting longstanding illnesses and mental health difficulties using cross-sectional studies in the UK. This study explored the association between having any limiting longstanding illness and serious psychological distress or of currently receiving treatment for depression or serious anxiety at age 17 years.

### Methods

A secondary analysis of the UK Millennium Cohort Study was conducted. Outcome measures were dichotomised responses from the self-administered Kessler-6 questionnaire for nonspecific psychological distress, and self-reported current treatment for depression or serious anxiety. Limiting longstanding illness data and covariates were taken from questionnaires with parents and adolescents, from birth up to age 17 years. Data were analysed using multiple binary logistic regression, first using complete-case analysis, and then using multiple imputation using chained equations.

### Results

Adolescents with a history of limiting longstanding illness were at an increased odds of both serious psychological distress (odds ratio = 1.53, 95%CI = 1.27–1.86) and self-reporting currently receiving treatment for depression or serious anxiety at age 17 years (odds ratio = 3.02, 95%CI 2.24–4.07).

**Data availability statement:** All relevant data for this study are publicly available from the UK Data Service repository (http://doi.org/10.5255/UKDA-Series-2000031).

**Funding:** G.P. is funded by a National Institute for Health and Care Research (NIHR) Pre-doctoral Fellowship (Ref: NIHR303343). This study has received funding from the National Lottery Community Fund (previously the Big Lottery Fund) as part of the A Better Start programme (Ref 10094849) and support from the NIHR Yorkshire and Humber Applied Research Collaboration (ARC-YH; Ref: NIHR200166, see https://www.arc-yh.nihr.ac.uk). L.F. is funded by a National Institute for Health and Care Research (NIHR) Career Development Fellowship (award: CDF-2018-11-ST2-002). S.J. was funded by an National Institute for Health and Care Research fellowship (DRF-2018-11-ST2-013). The views expressed are those of the authors and not necessarily those of the National Institute for Health and Care Research or the Department of Health and Social Care. The funders had no role in the study design, data collection and analysis, decision to publish, or preparation in the analysis.

**Competing interests:** The authors have declared that no competing interests exist.

## Conclusion

Children and young people with a limiting longstanding illness are at increased risk of having serious psychological distress, depression, and serious anxiety. Practitioners should be aware of this and routine screening for psychological distress, plus additional preventative support, may be beneficial.

## Introduction

Adolescence marks the period between childhood and adulthood, representing a key period in both physical and psychological development. National UK survey data found a prevalence of probable mental illness of 17.6% in those aged 6–16 years old in England [1], compared with 16.7% in adults in similar surveys [2]. Recently, mental health disorders for children and adolescents have become a public health crisis in England. Child referrals to mental health services more than doubled between 2020 and 2021 [3], and one in every five children aged 8–16 years old now experiences a mental health disorder [4]. A previous analysis of US national survey data [5] found that half of lifetime mental health disorders in the Diagnostic and Statistical Manual of Mental Disorders (DSM-IV) [6] start by 14 years of age, and three-quarters occur by 24 years. Adolescent mental illness is therefore a critical period for research. Mental illness experienced in adolescence is associated with significant potential harms including lower educational achievement [7], alcohol and substance misuse [8], social isolation [9], and suicide and self-inflicted injury [10].

Longstanding illness is defined by The UK Department of Health [11] as any illness or condition which cannot be cured, but can instead be treated with pharmacological or other treatments; it is limiting if it reduces the ability of someone to take part in normal activities. As medical care improves, childhood mortality rates in those aged 1–15 years in England and Wales have decreased from 33 deaths per 100,000 in 1981–8 deaths per 100,000 in 2021 [12]. This, amongst other factors including a rise in sedentary lifestyles, has resulted in a global rise in the proportion of those surviving to live with a limiting longstanding illness (LLSI) in all age groups, including children and adolescents [13]. Adolescents with a limiting longstanding illness (LLSI) may face difficulties such as hospital appointments, disruptions to their routine and education, and reduced capability to take part in regular activities; these challenges could lead to mental health difficulties [14]. It is important to understand any association between physical and mental illnesses to inform prevention strategies and paediatric service delivery, and to understand whether the early provision of psychological support may be necessary for all children and adolescents with a chronic illness.

Previous cross-sectional and longitudinal studies have identified associations between LLSIs and mental health difficulties in children outside of the UK [15–17]. Systematic reviews have identified that in adolescents, studies are largely constrained to identifying associations between diabetes or asthma and mental illness, with some evidence suggesting higher prevalence of mental health difficulties in those with inflammatory bowel disease, cystic fibrosis, and sickle cell disease

[18–20]. Further, few studies exploring longitudinal relationships between longstanding illnesses and mental health difficulties were at a low risk of methodological bias.

An analysis of a UK-based survey (n = 9,834) found that children and adolescents aged 5–15 years old with asthma were at a slightly increased odds of anxiety (OR=1.38, 95%CI 1.05–1.82) and hyperkinetic disorders (OR=1.64, 95%CI 1.00–2.39) when compared with those without asthma, measured using the Development and Well-Being Assessment [21]. Children with asthma who had parent-rated poor health were at significantly increased odds of scoring as borderline or abnormal for the parent (OR 3.1, 95%CI 2.4–4.1), teacher (OR 1.7, 95%CI 1.2–2.4), and child- (OR 1.9, 95%CI 1.2–3.0) rated total difficulties score on the Strengths and Difficulties Questionnaire, when compared with children without asthma who were in good health. Those with asthma but were in good health were not at a significantly increased odds of increased score on the Strengths and Difficulties Questionnaire, indicating that increased symptom severity in children with asthma may lead to increased risk of mental health difficulties.

One UK study (n = 14,775) using a secondary analysis of the Avon Longitudinal Study of Parents and Children found that those with a chronic illness had increased likelihood (OR 1.6, 95%CI 1.14–2.25) of having mental illness as measured by the Development and Well-Being Assessment at aged 15 years, compared with those without a chronic illness [22]. However, participants in this study were recruited from a single region in southwestern England, with those from disadvantaged or ethnic minority backgrounds being under-represented, and those from white ethnicity and advantaged backgrounds were over-represented.

As studies in adults have previously identified a bi-directional relationship between LLSI and mental health difficulties [23,24], this must be considered when interpreting estimates from cross-sectional studies. This study therefore explored the association between having any LLSI before aged 17 years with serious psychological distress at age 17 years in a large representative UK birth cohort- the Millennium Cohort Study. This study also investigated the relationship between LLSI and receiving treatment for depression or serious anxiety to explore not only any links between LLSI and psychological distress, but also associations with related healthcare use and to explore any differences between presence of psychological distress and access to related treatment.

## Methods

### Data source

This study uses a retrospective secondary analysis of data from the Millennium Cohort Study, a nationally representative UK birth cohort consisting of 19,517 cohort members and their families. Cohort members were recruited from the Child Benefit Register at age 9 months in data collection sweep 1 between the years 2000–2002, with some participants whose geographical data was unavailable recruited in sweep 2 at age 3 years [25]. A stratified sampling design was used, recruiting additional participants from areas with higher rates of minority ethnic groups and deprivation than the UK average. There are currently 7 sweeps of data; most recently when cohort members were aged 17 years [25]. Parents provided consent to take part in the study on behalf of cohort members up to age 11 years; from 14 years onwards, cohort members provided consent themselves.

### Outcome measures

Cohort members completed the Kessler-self-administered 6-item questionnaire (K6) at age 17 years. The scale is designed to measure non-specific psychological distress, and is scored from 0–24, with a cut-point of ≥13 indicating serious distress used frequently throughout the literature [26–28]. All participants with scores between 0–12 were coded as not having psychological distress, and participants with scores between 12–24 were considered as having serious psychological distress [28]. The K6 has an area under the receiver operating characteristic curve (AUROC) of 86.5%, compared with 85.4% in the 10-item version of the Kessler scale (K10), indicating the K6 performs slightly better than the

K10 at distinguishing between those with and without serious mental illness [29]. The scale demonstrates high internal consistency, with Cronbach's alpha estimates across varying languages, countries, and settings ranging from 0.81 to 0.89 in a single factor structure [29–32].

At age 17, cohort members were asked 'Has a doctor ever told you that you suffer from depression or serious anxiety?'. Where cohort members responded with 'yes' they were further asked 'are you currently being treated for depression or serious anxiety?'. Those who answered 'yes' to both questions were counted as having the second outcome of current treatment for depression or serious anxiety. This enabled some exploration of contact with healthcare services and treatment.

### Exposure

LLSI status was gathered on behalf of cohort members from parent/main carer interviews, between the ages of 3 and 14 years. The parent or main carer of each cohort member was asked 'Does [cohort member] have a longstanding illness or condition?' at each sweep; for those who answered 'yes' they were further asked 'Does this illness limit [cohort member]'s activity in any way?'. For the purposes of this study, respondents whose parent or main carer answered 'yes' to both questions at any sweep were considered to have an LLSI, and respondents whose parents answered no to both, or no to the second question were considered to not have an LLSI, and were used as the reference group. At age 11 and 14 years, participants who answered 'yes' were also asked which health conditions affect the cohort member; those who reported being affected exclusively by a mental health condition, and not by any other condition, were not considered to have an LLSI.

### Covariates

All covariates were taken from face-to-face interviews with each cohort member's parent or main carer, at age 9 months for those in the original sample, and age 3 years for those in the booster sample. Covariates selection for this study was informed by previous studies; all covariates, are theorised to confound the relationship between LLSI and serious psychological distress [33–36] [see Fig 1]. Covariates were:

- cohort member sex (coded as 0 = male, 1 = female)

- ethnic group (coded as 0 = White British (reference group), 1 = Pakistani and Bangladeshi, 2 = Black or Black British, 3 = Mixed or Other Ethnic group)

- family structure (coded as 0 = dual-parent family, 1 = single-parent family)

- being above or below the 60% median on the Organisation for Economic Co-operation and Development equivalised income scale (coded as 0=above median, 1=below median)

- highest parental qualification, coded according to National Vocational Qualification (NVQ) or equivalent levels, i.e., (as 0 = none or overseas qualification, 1 = NVQ level 1 (lower grades at school qualifications typically gained at age 16), 2 = NVQ level 2 (higher grades at school qualification typically gained at age 16, 3 = NVQ level 3 (school qualifications typically gained at age 18), 4 = NVQ level 4 (further education or undergraduate degree level qualifications), 5 = NVQ level 5 (higher than degree – e.g., masters level/doctoral – qualifications)

- parental self-reported LLSI (coded as 0 = no LLSI, 1 = LLSI present)

- parental K6 score (continuous).

### Statistical analyses

For the primary outcome of psychological distress measured by K6 score, complete-case analyses were used. Complete case analyses were also used for the secondary outcome of whether a cohort member reported currently receiving

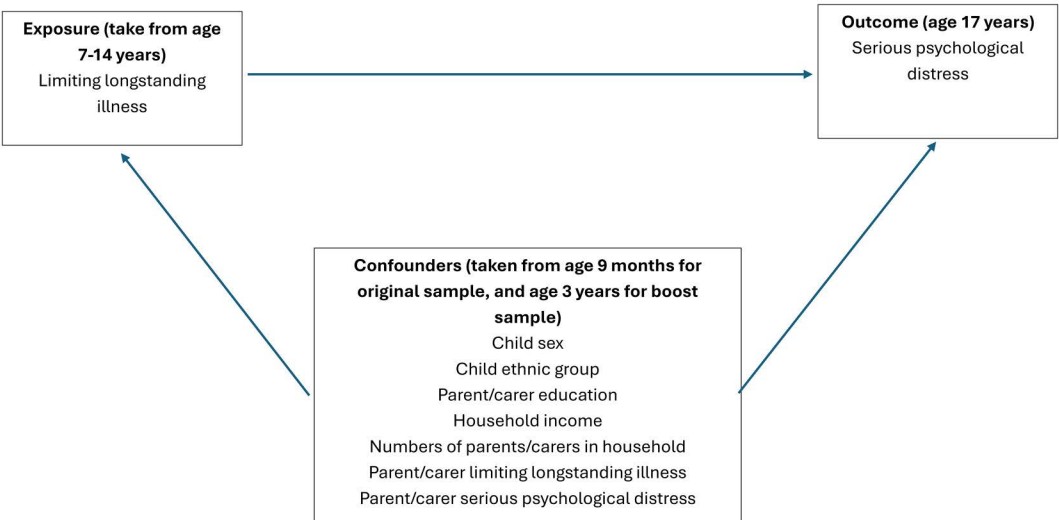

**Fig 1. Directed acyclic graph of relationships between measured variables in this study.** Note: arrows between confounders have been removed to simplify the presentation of the DAG.

treatment for depression or serious anxiety. To minimise attrition, participants with LLSI data for at least one sweep were included, providing that all covariate data were available. Where there was more than one cohort member in a family, the second cohort member was excluded to prevent clustering by household. The MCS used a stratified sampling design, over-sampling from electoral wards with higher levels of ethnic diversity and socioeconomic deprivation in the 1991 UK Census [37]. Sampling and response weights were used to account for both sampling design and attrition in all analyses.

Binary logistic regression was used to produce crude and adjusted odds ratios (ORs) for the association between LLSI and dichotomised K6 score, and between LLSI and receiving treatment for depression or serious anxiety. Two-sided statistical tests using a p-value threshold of <0.05 were used throughout. All covariates listed above were included in adjusted models for both outcomes, using multiple logistic regression, which were compared with unadjusted ORs from models using simple binary logistic regression. Additionally, a sensitivity analysis for the model used for the primary outcome of K6 score was also conducted using multiple imputation using chained equations (MICE) to impute missing data for all variables included in the logistic regression model investigating the relationship between LLSI and K6 score (see Fig 1 for numbers of missing values). All variables used in the analysis model for the primary outcome were included in the imputation model. The Stata command 'mi impute chained' was used to impute 20 datasets. This was used to assess how the use of complete-case analyses influenced model estimates. All analyses were conducted in Stata Version 17.

## Results

Of the 19,482 cohort members in end-user license agreement dataset, 7,866 cohort members provide complete exposure, outcome, and covariate data, representing 40.4% of the original sample (Fig 2).

Unweighted cohort member characteristics for the complete-case dataset are presented in Table 1; weighted figures are available in Appendix S1 in S1 File. 15.8% of cohort members scored above the threshold for serious distress on the K6, with 21.9% of participants who reported LLSI before age 17 years scoring above the cutoff, compared with 14.5% in the unexposed group. Greater proportional differences were seen in receiving treatment for depression or serious anxiety; 9.7% in those with reported LLSI, compared to 3.5% in those without. Males had higher prevalence of LLSI, with 53.6% of those reporting an LLSI being male. Those with a reported LLSI also saw slightly higher prevalence of LLSI in proportions

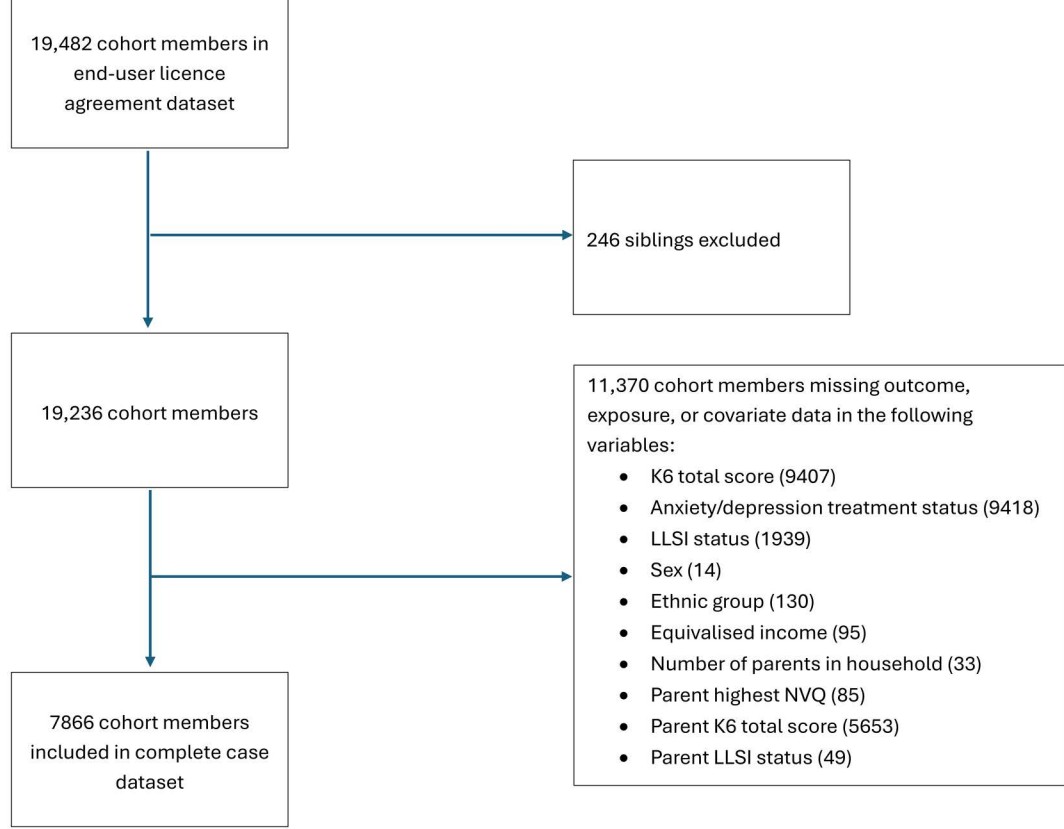

**Fig 2. Attrition diagram showing those included in complete-case analyses.**

of those from single-parent families, those below the 60% median equivalised household income threshold, and those from the White ethnic group, although the study population is also overwhelmingly white at 88.1%.

### LLSI and serious psychological distress

21.9% (302) of those with an LLSI scored above the threshold for serious psychological distress on the Kessler-6 compared with 14.5% (938) of those without an LLSI. The crude OR for serious psychological distress at age 17 years in those with LLSI compared to those without was 1.53 (95%CI 1.29–1.83). The weighted adjusted OR was also 1.53 (95%CI 1.27–1.86, Table 2). Unweighted complete-case analyses can be seen in Appendix S2 in S1 File. Models using MICE to account for missing data retained 19,229 cohort members and produced an adjusted OR of 1.58 (95%CI 1.36–1.84), which was similar to the estimates from complete-case analyses (Appendix S3 in S1 File).

### LLSI and treatment for depression or serious anxiety

9.7% (134) of those with an LLSI reported currently receiving treatment for depression or serious anxiety, compared with just 3.5% (225) in those without an LLSI. The crude OR for children and adolescents who reported currently receiving treatment for depression or serious anxiety by age 17 years in those with an LLSI compared to those without was 3.18 (95%CI 2.40–4.22), and the adjusted value was 3.02 (95%CI 2.24–4.07, Table 3). Unweighted complete-case analyses can be seen in Appendix S4 in S1 File.

**Table 1. Unweighted summary statistics of covariates by LLSI status in complete case dataset.**

| Variable | No LLSI (%) | LLSI (%) | Total (%) |
|---|---|---|---|
| **Categorical variables** | | | |
| Sex | | | |
| Female | 3,369 (51.9) | 640 (46.4) | 4,009 (51.0) |
| Male | 3,119 (48.1) | 738 (53.6) | 3,857 (49.0) |
| **Ethnic group** | | | |
| White | 5,696 (87.8) | 1,227 (89.0) | 6,923 (88.0) |
| Indian | 135 (2.1) | 28 (2.0) | 163 (2.0) |
| Pakistani and Bangladeshi | 267 (4.1) | 50 (3.6) | 317 (4.0) |
| Black or Black British | 159 (2.5) | 29 (2.1) | 188 (2.4) |
| Mixed or Other Ethnic group | 231 (3.6) | 44 (3.2) | 275 (3.5) |
| **Family structure** | | | |
| Two parents/carers | 5,769 (88.9) | 1,160 (84.2) | 6,929 (88.1) |
| Single parent/carer | 719 (11.1) | 218 (15.8) | 937 (11.9) |
| **Parental education** | | | |
| None or overseas qualification | 642 (9.9) | 157 (11.4) | 799 (10.2) |
| NVQ level 1 | 459 (7.1) | 114 (8.3) | 573 (7.3) |
| NVQ level 2 | 1,825 (28.1) | 417 (30.3) | 2,242 (28.5) |
| NVQ level 3 | 990 (15.3) | 212 (15.4) | 1,202 (15.3) |
| NVQ level 4 | 2,258 (34.8) | 423 (30.7) | 2,681 (34.1) |
| NVQ level 5 | 314 (4.8) | 55 (4.0) | 369 (4.7) |
| **Parental limiting longstanding illness** | | | |
| No | 5,217 (80.4) | 967 (70.2) | 6,184 (78.6) |
| Yes | 1,271 (19.6) | 411 (29.8) | 1,682 (21.4) |
| **OECD equivalised income** | | | |
| Above 60% median | 4,879 (75.2) | 909 (66.0) | 5,788 (73.6) |
| Below 60% median | 1,609 (24.8) | 469 (34.0) | 2,078 (26.4) |
| **Continuous variables** | **No LLSI** mean (SD) | **LLSI** mean (SD) | **Total** mean (SD) |
| **Parental Kessler-6 score** | 2.96 (3.40) | 3.93 (4.20) | 3.13 (3.57) |

## Discussion

This was the first UK-wide longitudinal study investigating the relationship between LLSI and psychological distress. This study found that children and adolescents with a known history of LLSI had an odds of 1.53 having serious psychological distress at age 17, when compared with those without an LLSI. Those with a known history of LLSI had an odds of 3.02 for reporting currently receiving treatment for depression or serious anxiety by a healthcare professional at age 17 years, when compared with those without an LLSI.

The weighted prevalence of LLSI in this study of 15.7% (Appendix S1 in S1 File) is greater than that seen in national survey data for children aged 0–15 years in a comparable time period such as The Health Survey for England at 8% [38]. The use of a longitudinal measure of LLSI in this study, which placed those with any history of self-reporting an LLSI in the exposure group may be responsible for some of this difference. Varying age ranges, areas, and definitions of LLSI across studies also make between-studies comparison of LLSI prevalence difficult, with other England surveys showing higher rates at 23% [39].

**Table 2. Weighted complete-case multiple logistic regression of dichotomised Kessler-6 score.**

| Covariate | Weighted adjusted OR | 95% CI | P-value |
|---|---|---|---|
| **Limiting longstanding illness** | | | |
| No limiting longstanding illness | 1.00 | | (ref) |
| Limiting longstanding illness | 1.53 | 1.27-1.86 | <0.001 |
| **Sex** | | | |
| Female | 1.00 | | |
| Male | 0.37 | 0.31-0.45 | <0.001 |
| **Ethnic group** | | | |
| White | 1.00 | | (ref) |
| Indian | 0.43 | 0.22-0.84 | 0.013 |
| Pakistani/ Bangladeshi | 0.59 | 0.37-0.94 | 0.025 |
| Black/ Black British | 1.07 | 0.51-2.24 | 0.854 |
| Mixed/ Other | 0.89 | 0.52-1.52 | 0.670 |
| **Family structure** | | | |
| Dual parents/carers | 1.00 | | (ref) |
| Single parent/carer | 1.10 | 0.61-1.49 | 0.561 |
| **Parental education** | | | |
| None or overseas only | 1.00 | | (ref) |
| NVQ level 1 | 1.02 | 0.61-1.70 | 0.942 |
| NVQ level 2 | 0.99 | 0.71-1.40 | 0.968 |
| NVQ level 3 | 1.00 | 0.69-1.47 | 0.983 |
| NVQ level 4 | 0.94 | 0.66-1.33 | 0.733 |
| NVQ level 5 | 0.94 | 0.60-1.34 | 0.779 |
| **OECD equivalised income** | | | |
| Above 60% median | 1.00 | | (ref) |
| Below 60% median | 1.17 | 0.97-1.41 | 0.108 |
| **Parental limiting longstanding illness** | | | |
| No | 1.00 | | (ref) |
| Yes | 1.08 | 0.89-1.30 | 0.423 |
| **Parental Kessler-6 score** | 1.04 | 1.02-1.07 | <0.001 |
| **Constant** | 0.22 | 0.15-0.31 | <0.001 |

*Cutoff Kessler-6 score of ≥13 used to indicate serious psychological distress*

All LLSI data were taken before both outcome measurements at age 17 years. Those who reported having an LLSI at age 11–14 years, but self-categorised this as only having a mental health condition with no other type of LLSI were excluded from the exposure group for the purposes of this study. However, some participants were missing data at these sweeps, so it is not possible to assess the mental health status of some participants between the ages of 11–14 years. The longitudinal design of the current study therefore does facilitate some causal interpretation to indicate that LLSI may increase the risk of serious psychological distress, and of receiving treatment for depression or serious anxiety at age 17 years. Mediation analyses and assessment of dose-response relationships, such as severity or duration of LLSI in other populations could further explore this.

This study identified a greater proportion of adolescents with recent serious nonspecific psychological distress (15.8%) than those who were undergoing treatment for depression or serious anxiety (4.6%). This could indicate underdiagnosis

**Table 3. Weighted complete-case logistic regression of currently receiving treatment for depression or serious anxiety.**

| Covariate | Weighted adjusted OR | 95% CI | P-value |
|---|---|---|---|
| **Limiting longstanding illness** | | | |
| No limiting longstanding illness | 1.00 | | (ref) |
| Limiting longstanding illness | 3.02 | 2.24-4.07 | <0.001 |
| **Sex** | | | |
| Female | 1.00 | | |
| Male | 0.29 | 0.21-0.39 | <0.001 |
| **Ethnic group** | | | |
| White | 1.00 | | (ref) |
| Indian | 0.67 | 0.16-2.71 | 0.572 |
| Pakistani/ Bangladeshi | 0.32 | 0.12-0.86 | 0.025 |
| Black/ Black British | 0.48 | 0.14-1.64 | 0.241 |
| Mixed/ Other | 0.22 | 0.07-0.70 | 0.010 |
| **Family structure** | | | |
| Dual parents/carers | 1.00 | | (ref) |
| Single parent/carer | 0.79 | 0.46-1.36 | 0.398 |
| **Parental education** | | | |
| None or overseas only | 1.00 | | (ref) |
| NVQ level 1 | 1.28 | 0.69-2.38 | 0.435 |
| NVQ level 2 | 0.80 | 0.47-1.33 | 0.384 |
| NVQ level 3 | 0.61 | 0.35-1.08 | 0.087 |
| NVQ level 4 | 0.71 | 0.43-1.17 | 0.173 |
| NVQ level 5 | 0.97 | 0.49-1.90 | 0.921 |
| **OECD equivalised income** | | | |
| Above 60% median | 1.00 | | (ref) |
| Below 60% median | 1.09 | 0.76-1.55 | 0.646 |
| **Parental limiting longstanding illness** | | | |
| No | 1.00 | | (ref) |
| Yes | 1.35 | 0.98-1.86 | 0.069 |
| **Parental Kessler-6 score** | 1.06 | 1.02-1.09 | 0.003 |
| **Constant** | 0.06 | 0.04-0.09 | <0.001 |

of depression and serious anxiety, although some cohort members could have experienced psychological distress without meeting the threshold for treatment for depression or serious anxiety. It is important for healthcare practitioners to be aware of the increased psychological distress in adolescents with an LLSI. A separate pilot study implemented routine screening and an early stepped intervention pathway into a paediatric epilepsy clinic and was able to identify children with previously unidentified mental health difficulties and enable them to appropriately access care [40]. In another study, routine depression screening in adolescents with type 1 diabetes mellitus at quarterly appointments in a Cincinnati Children's Hospital was able to detect depression symptoms early and create opportunities for early intervention [41]. As the current study identified increased odds of serious psychological distress or of receiving treatment for depression or serious anxiety, it may be beneficial to screen for psychological distress and provide early interventions more broadly in adolescents with any LLSI, to facilitate early detection of mental health difficulties. Future studies could investigate the relationship between contact with health professionals and diagnosis and treatment of mental health difficulties to understand this further, and to improve access to mental health support in young people.

 

The OR for receiving treatment for serious depression or anxiety (3.02) was greater than when measuring serious psychological distress using the K6 (1.53). As LLSIs often require treatment or regular monitoring, adolescents with an LLSI may have increased contact with healthcare professionals when compared with their peers. This may lead to increased opportunity for healthcare professionals to identify mental health difficulties in children and young people with LLSIs. It is possible that those without LLSIs may constitute a larger proportion of those with undiagnosed mental health difficulties. Further exploration of the potential benefits of screening for psychological distress outside of those with an LLSI may also be useful for the early identification of mental health difficulties.

This study adjusted for the effects of ethnicity in all adjusted models, however, the representativeness of some ethnic minority groups in the Millennium Cohort Study is limited. Previous literature has shown that young people from ethnic minority and socioeconomically deprived backgrounds face substantial barriers to accessing quality support to manage their physical and mental health [42]. Future research could examine whether the association found here varies across different ethnic groups, to establish whether the association may be stronger for groups who have more difficulties with managing their mental health.

The increased odds of serious psychological distress in those with an LLSI in this study is comparable to odds ratios from the previously mentioned UK studies investigating associations between chronic illness and mental illness (1.6) [22] and asthma and mental illness (1.38) [21]. The consistency of results suggests that the risk of mental illness may be higher for children with any LLSI. However, the added benefit of this study was the use of a longitudinal design in a more representative UK sample, covering a broader range of LLSIs than those seen in previous studies.

Although this study has found increased odds of mental health difficulties in adolescents with an LLSI, it is important to consider the moderating and potentially neutralising effects of family support, education, and management of the physical condition itself before considering mental health interventions. Adolescence often marks the transition from parent- to self-management of LLSI, with differences in the age this occurs seen across individuals and conditions [43]. As a result of this and other developmental and environmental changes, non-adherence to treatments across illnesses including asthma [44] and diabetes [45] is particularly high in adolescents. This is concerning, as non-adherence with treatment of LLSI may exacerbate symptoms or illness progression; it is possible that this could also have adverse effects on mental health outcomes. The reverse may also be true – a systematic review identified that adolescents self-reported that poor mental health made it more difficult to adhere to treatment for physical illnesses [46]. Few studies appear to have examined the mental health implications of improving LLSI treatment adherence in adolescents. As smartphone use becomes increasingly popular, one study found that a Mobile-health app which included symptom monitoring questionnaires, medication reminders, information videos, and capacity to talk to pharmacists or peers improved treatment adherence in adolescents with a low adherence at baseline [47]. A trial found that adolescents with type 1 diabetes who had received motivational interviewing had improved glycaemic control and life satisfaction, and had reduced worry [48]. Future studies could further explore the mental health impacts of comparable interventions and alternative approaches such as coaching across other chronic conditions.

## Strengths and limitations

This study has several strengths. The Millennium Cohort Study provided a large nationally representative UK sample, with a prospective study design which reduced recall bias compared to a single cross-sectional analysis, and sufficient demographic and socioeconomic information for use as confounding variables.

This study also has some limitations. The high attrition rate and use of complete case analysis in this study meant that 41% of singleton cohort members recruited to the Millennium Cohort Study were included in the complete-case analyses. This may reduce the representativeness of results and lead to attrition bias. Although sensitivity analyses using multiple imputation by chained equations to account for missingness did not lead to substantial changes in model estimates, if the missing at random assumption does not hold, then estimates using multiple imputation are also likely to be biased [49].

Prior to age 11 years, parents and carers were not asked what type of health condition cohort members were affected by. This may have lead to some cohort members with a limiting mental health condition prior and no LLSI prior to age 11 years being misclassified as having an LLSI, which may have inflated the relationship between LLSI and serious psychological distress in this study.

## Conclusion

This was the first large-scale longitudinal study to investigate the relationship between LLSI and psychological distress in a representative sample of UK adolescents. This study found that children and adolescents with a known LLSI are at an increased risk of serious psychological distress and of currently receiving treatment for depression or serious anxiety at age 17 years when compared with those without an LLSI. Working alongside young people to provide support, improve treatment adherence, and routinely screen for and identify psychological distress in young people with any LLSI may be beneficial.

## Supporting information

**S1 File. Supplementary appendices.** Supplementary appendices containing (S1) Weighted summary statistics of covariates by LLSI status in complete case dataset; (S2) Unweighted complete-case multiple logistic regression for dichotomised Kessler-6 score; (S3) Weighted multiple logistic regression of dichotomised Kessler-6 score using Multiple Imputation by Chained Equations; (S4) Unweighted complete-case multiple logistic regression of currently receiving treatment for depression or serious anxiety.
(DOCX)

## Author contributions

**Conceptualization:** Gareth Martyn Palliser, Lorna K. Fraser, Stuart W. Jarvis.

**Data curation:** Gareth Martyn Palliser, Stuart W. Jarvis.

**Formal analysis:** Gareth Martyn Palliser, Kate E. Mooney, Stuart W. Jarvis.

**Investigation:** Gareth Martyn Palliser.

**Methodology:** Gareth Martyn Palliser, Lorna K. Fraser, Kate E. Mooney, Stuart W. Jarvis.

**Project administration:** Gareth Martyn Palliser.

**Supervision:** Lorna K. Fraser, Stuart W. Jarvis.

**Validation:** Gareth Martyn Palliser, Kate E. Mooney, Stuart W. Jarvis.

**Visualization:** Gareth Martyn Palliser.

**Writing – original draft:** Gareth Martyn Palliser, Stuart W. Jarvis.

**Writing – review & editing:** Gareth Martyn Palliser, Lorna K. Fraser, Kate E. Mooney, Stuart W. Jarvis.

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
