## [Decision Letter · Decision Letter 0]

19 Feb 2025

PONE-D-24-25246The association between limiting longstanding illness and serious psychological distress in adolescents: A secondary analysis of the UK Millennium Cohort StudyPLOS ONE

Dear Dr. Jarvis,

Thank you for submitting your manuscript to PLOS ONE. After careful consideration, we feel that it has merit but does not fully meet PLOS ONE’s publication criteria as it currently stands. Therefore, we invite you to submit a revised version of the manuscript that addresses the points raised during the review process.

We look forward to receiving your revised manuscript.

Kind regards,

Shivanand Kattimani

Academic Editor

PLOS ONE

Reviewers' comments:

Reviewer's Responses to Questions

**Comments to the Author**

1. Is the manuscript technically sound, and do the data support the conclusions?

Reviewer #1: Partly

Reviewer #2: Partly

2. Has the statistical analysis been performed appropriately and rigorously? 

Reviewer #1: Yes

Reviewer #2: Yes

3. Have the authors made all data underlying the findings in their manuscript fully available?

Reviewer #1: Yes

Reviewer #2: Yes

4. Is the manuscript presented in an intelligible fashion and written in standard English?

Reviewer #1: Yes

Reviewer #2: No

5. Review Comments to the Author

Reviewer #1: Dear Authors,

I appreciate the opportunity to review your manuscript, which presents an important examination of LLSI and psychological distress at age 17. I found the paper compelling, but I have a few suggestions that I believe would strengthen the clarity and flow of the manuscript.

In Line 29, the phrase “self-reported currently receiving treatment for depression or serious anxiety” felt a bit clunky and disrupted the flow of the text. I understand the intent behind capturing the self-reported aspect of treatment, but I wonder if there might be a more concise way to phrase this.

I also noticed areas where the flow could be improved by adjusting the placement of citations or statistics. For example, in Line 94, shifting the reported odds to follow "increased odds" might make the sentence easier to read. Similarly, I observed that citations are often placed at the beginning of sentences — in some cases, it might be more helpful to cite studies closer to the outcome statement, especially when referencing multiple sources. That said, I recognize that these are stylistic preferences, so please feel free to disregard if not relevant to your chosen writing style.

One area where I think the manuscript could benefit from further development is in the introduction. While you make a strong case for examining LLSI and distress, the importance of exploring whether treatment was being received is less clearly articulated. Expanding on this point would help reinforce the rationale for including treatment status as a key variable.

I also had some difficulty understanding the definition and classification of LLSI, and there seemed to be some discrepancies from the introduction to the methods section. In Line 134, the statement that participants answering "yes" to both questions were considered to have an LLSI seemed to suggest that receiving treatment might be a criterion for LLSI. This was a bit confusing, as I had understood LLSI to refer to chronic illness limiting activities, with treatment status as a separate variable so that associations could be examined. If treatment is not a criterion for LLSI, clarifying this distinction in the methods section would be helpful to avoid misinterpretation.

More broadly, I would encourage revisiting the definitions of LLSI and the measured variables to ensure readers can clearly understand how they are operationalized. As it currently reads, there’s a risk of confounding the association between chronic illness and distress if mental health diagnosis or treatment is part of the LLSI classification, which would be concerning from a validity of outcomes perspective. If this isn’t the case, tightening the definitions and explicitly outlining the variables measured would address this concern.

Overall, I think this is a valuable piece of research, and with some adjustments to definitions, structure, and flow, it will be even stronger.

Reviewer #2: Dear Editor, the manuscript is basically good.

In #4 above I selected "no" when asked about the writing standard. In fact, the authors do write well, but there are areas where they can explain their purpose more clearly. For example, they use a definition of LLSI (limiting and long-lasting illness) in different ways from the originally presented definition.

Additionally, as the data base is from a large UK study, the authors refer to UK codes that represent, as an example, various work-level advances (NVK). These are not explained, and readers outside of UK will not likely know what this refers to. So my primary concern as that the paper at times does not clearly explain some of it's intentions, and does not always explain terminologies.

The comments ask for basic changes. Overall it is a good manuscript from my perspective.

Thank you for the opportunity to review this manuscript!

6. PLOS authors have the option to publish the peer review history of their article (what does this mean? ). If published, this will include your full peer review and any attached files.

**Do you want your identity to be public for this peer review?** For information about this choice, including consent withdrawal, please see our Privacy Policy .

Reviewer #1: No

Reviewer #2: No

---

## [Author Response · Author response to Decision Letter 1]

21 Jul 2025

The response to reviewers is included as a separate word document.

---

## [Decision Letter · Decision Letter 1]

14 Aug 2025

The association between limiting longstanding illness and serious psychological distress in adolescents: A secondary analysis of the UK Millennium Cohort Study

PONE-D-24-25246R1

Dear Dr. Jarvis,

We’re pleased to inform you that your manuscript has been judged scientifically suitable for publication and will be formally accepted for publication once it meets all outstanding technical requirements.

Kind regards,

Shivanand Kattimani

Academic Editor

PLOS ONE

Reviewers' comments:

Reviewer's Responses to Questions

**Comments to the Author**

1. If the authors have adequately addressed your comments raised in a previous round of review and you feel that this manuscript is now acceptable for publication, you may indicate that here to bypass the “Comments to the Author” section, enter your conflict of interest statement in the “Confidential to Editor” section, and submit your "Accept" recommendation.

Reviewer #1: All comments have been addressed

Reviewer #2: All comments have been addressed

2. Is the manuscript technically sound, and do the data support the conclusions?

Reviewer #1: (No Response)

Reviewer #2: Yes

3. Has the statistical analysis been performed appropriately and rigorously? 

Reviewer #1: (No Response)

Reviewer #2: Yes

4. Have the authors made all data underlying the findings in their manuscript fully available?

Reviewer #1: (No Response)

Reviewer #2: Yes

5. Is the manuscript presented in an intelligible fashion and written in standard English?

Reviewer #1: (No Response)

Reviewer #2: Yes

6. Review Comments to the Author

Reviewer #1: (No Response)

Reviewer #2: This topic is very interesting and important. Thanks to the authors for improving the manuscript based upon recommendations from the reviewers. The manuscript reads well and will contribute meaningful information to the scientific literature!

7. PLOS authors have the option to publish the peer review history of their article (what does this mean? ). If published, this will include your full peer review and any attached files.

**Do you want your identity to be public for this peer review?** For information about this choice, including consent withdrawal, please see our Privacy Policy .

Reviewer #1: No

Reviewer #2: No

---

## [Editor Report · Acceptance letter]

PONE-D-24-25246R1

PLOS ONE

Dear Dr. Jarvis,

I'm pleased to inform you that your manuscript has been deemed suitable for publication in PLOS ONE. Congratulations! Your manuscript is now being handed over to our production team.

Kind regards,

on behalf of

Dr. Shivanand Kattimani

Academic Editor

PLOS ONE